# Monocyte Subsets in Patients with Chronic Heart Failure Treated with Cardiac Resynchronization Therapy

**DOI:** 10.3390/cells10123482

**Published:** 2021-12-09

**Authors:** Katarzyna Ptaszyńska-Kopczyńska, Andrzej Eljaszewicz, Marta Marcinkiewicz-Siemion, Emilia Sawicka-Śmiarowska, Ewa Tarasiuk, Anna Lisowska, Marlena Tynecka, Kamil Grubczak, Urszula Radzikowska, Adrian Janucik, Marcin Moniuszko, Karol Charkiewicz, Piotr Laudański, Bożena Sobkowicz, Karol A. Kamiński

**Affiliations:** 1Department of Cardiology, Medical University of Białystok, ul. Skłodowskiej-Curie 24A, 15-089 Białystok, Poland; kasia.ptaszynska@op.pl (K.P.-K.); marcinkiewicz.m22@gmail.com (M.M.-S.); emiliasawickak@gmail.com (E.S.-Ś.); ewa-tarasiuk@o2.pl (E.T.); anlila@poczta.onet.pl (A.L.); sobkowic@wp.pl (B.S.); 2Department of Regenerative Medicine and Immune Regulation, Medical University of Białystok, ul. Waszyngtona 13, 15-269 Białystok, Poland; Andrzej.Eljaszewicz@umb.edu.pl (A.E.); marlena.tynecka@umb.edu.pl (M.T.); kamil.grubczak@umb.edu.pl (K.G.); urszula.radzikowska@umb.edu.pl (U.R.); adrian.janucik@umb.edu.pl (A.J.); Marcin.Moniuszko@umb.edu.pl (M.M.); 3Department of Allergology and Internal Medicine, Medical University of Białystok, ul. Skłodowskiej-Curie 24A, 15-276 Białystok, Poland; 4Department of Perinatology and Obstetrics, Medical University of Białystok, ul. Skłodowskiej-Curie 24A, 15-276 Białystok, Poland; karol.charkiewicz@abbvie.com (K.C.); piotr.laudanski@wum.edu.pl (P.L.); 51st Department of Obstetrics and Gynecology, Medical University of Warsaw, pl. S. Starynkiewicza 1/3, 02-015 Warsaw, Poland; 6Department of Population Medicine and Civilization Disease Prevention, Medical University of Białystok, ul. Waszyngtona 13, 15-269 Białystok, Poland

**Keywords:** cardiac resynchronization therapy, chemokines, heart failure, iron homeostasis, monocytes

## Abstract

Background: The exact role of individual inflammatory factor in heart failure with reduced ejection fraction (HFrEF) remains elusive. The study aimed to evaluate three monocyte subsets (classical-CD14^++^CD16^−^, intermediate-CD14^++^CD16^+^, and nonclassical-CD14^+^CD16^++^) in HFrEF patients and to assess the effect of the cardiac resynchronization therapy (CRT) on the changes in monocyte compartment. Methods: The study included 85 patients with stable HFrEF. Twenty-five of them underwent CRT device implantation with subsequent 6-month assessment. The control group consisted of 23 volunteers without HFrEF. Results: The analysis revealed that frequencies of non-classical-CD14^+^CD16^++^ monocytes were lower in HFrEF patients compared to the control group (6.98 IQR: 4.95–8.65 vs. 8.37 IQR: 6.47–9.94; *p* = 0.021), while CD14^++^CD16^+^ and CD14^++^CD16^−^ did not differ. The analysis effect of CRT on the frequency of analysed monocyte subsets 6 months after CRT device implantation showed a significant increase in CD14^+^CD16^++^ (from 7 IQR: 4.5–8.4 to 7.9 IQR: 6.5–9.5; *p* = 0.042) and CD14^++^CD16^+^ (from 5.1 IQR: 3.7–6.5 to 6.8 IQR: 5.4–7.4; *p* = 0.017) monocytes, while the frequency of steady-state CD14^++^CD16^−^ monocytes was decreased (from 81.4 IQR: 78–86.2 to 78.2 IQR: 76.1–81.7; *p* = 0.003). Conclusions: HFrEF patients present altered monocyte composition. CRT-related changes in the monocyte compartment achieve levels observed in controls without HFrEF.

## 1. Introduction

Heart failure with reduced ejection fraction (HFrEF) is a common condition with high morbidity and mortality, usually developing as a complication of the coronary artery disease or cardiomyopathies [1], Cardiac resynchronization therapy (CRT) is therapeutic method for HFrEF patients that improves the quality of life, functional status as well as prognosis, and provides the unique opportunity to obtain insights into the pathomechanisms underlying myocardial remodelling and development of HF [2].

One of the crucial mechanisms contributing to HF onset and progression is inflammatory activation. To date, however, the role of the systemic inflammatory and immunological processes in the development and progression of HFrEF remains elusive [3]. Chronic low-grade inflammation, reflected in overexpression of proinflammatory cytokines, such as interleukin-6 (IL-6), tumor necrosis factor-like weak inducer of apoptosis (TWEAK), and chemokines, has been recognized as the underlying pathomechanism of diseases affecting the cardiovascular system [4].

Monocytes represent integral and fundamental components of inflammatory and innate immune responses in the cardiovascular system [5]. Due to their pleiotropic biological activities, they may play both immune-stimulatory and regulatory roles. Notably, monocytes’ function is directly associated with their polarisation towards proinflammatory or anti-inflammatory cells. The differences include variability of phagocytosis, cytotoxicity, cytokine production, antigen presentation and angiogenic properties. Thus, monocytes can be beneficial or detrimental to the development of cardiovascular diseases, depending on systemic and local signalling mediated by soluble factors, including growth factors, chemokines and cytokines [6]. Currently, based on different expression of CD14 (coreceptor for lipopolysaccharide) and CD16 (Fc gamma III receptor) receptors, monocytes are divided into three functionally distinct subsets of monocytes, namely, classical (CD14^++^CD16^−^), intermediate (CD14^++^CD16^+^) and nonclassical (CD14^+^CD16^++^) [7]. The classical subset represents the majority of peripheral blood monocytes (up to 90%) acting as scavengers for apoptotic cells and microbial pathogens. The remaining two subsets, i.e., intermediate and nonclassical, are recognized as activated monocytes, and in several pathological conditions they were found to be similar to M2 (alternatively activated) and M1 (classically activated) macrophages, respectively [8].

Intermediate monocytes were found to be associated with an increase in cardiovascular events in patients with the chronic kidney disease [9]. Importantly, proinflammatory effects of the classical CD14^++^CD16^−^ monocytes may be compensated by beneficial properties of the CD14^++^CD16^+^ intermediate subset, including production of interleukin 10 (IL-10), stimulation of angiogenesis and tissue repair [10]. Notably, monocyte differentiation into pro- or anti-inflammatory cells is orchestrated by inflammatory mediators released by inflamed or injured tissues [10]. Although monocytes play an important role in the cardiovascular diseases, only a limited number of studies describe the monocyte profile in HF. Furthermore, to date, a potential effect of CRT-related reversal of myocardial remodelling on monocyte composition was not elucidated. Previous reports of our and other groups showed, however, that CRT is associated with dynamic changes in the systemic inflammatory mediator profiles [11,12].

Therefore, in this study, we aimed to evaluate the composition of the three major monocyte subsets, namely CD14^++^CD16^−^ classical, CD14^++^CD16^+^ intermediate, and CD14^+^CD16^++^ nonclassical, in the HFrEF patients. For better understanding of inflammatory activation, we analysed the association of the different monocyte subsets with selected cytokines and their soluble receptors, i.e., IL-6 and soluble IL-6 receptors (sIL-6R), soluble form of gp130 (sgp130), TNF-like weak inducer of apoptosis (TWEAK) and soluble decoy receptor [13] CD163 (sCD163), as well as 40 chemokines. Finally, we evaluated dynamic changes in different monocyte subsets in the HFrEF patients treated with CRT in relation to iron homeostasis.

## 2. Materials and Methods

### 2.1. Study Population

We enrolled 85 patients with stable, optimally pharmacologically treated chronic HF in NYHA class ≥II with left ventricular ejection fraction lower than 35% (HFrEF group) diagnosed in echocardiography. The aetiology of HF included dilated (34 (40%) patients) or ischemic cardiomyopathy (51 (60%) patients). Twenty-five patients from the HFrEF group, who met the criteria for resynchronization therapy, further underwent CRT device implantation.

The control group consisted of 23 volunteers (outpatients) without history of HF, with similar median age and body weight, affected with chronic cardiovascular diseases observed in the HFrEF patients (Table 1).

All the study participants underwent the same diagnostic assessment: medical interview with estimation of NYHA functional class, physical examination, transthoracic echocardiography, cardiopulmonary exercise test (CPET), 6 min walk distance (6MWD) test and venous blood tests. Transferrin saturation was obtained from the ratio of venous iron concentration to total iron binding capacity. Patients with active infection, neoplasm diagnosed within the last 5 years or significant chronic pulmonary disease, were excluded from the study. The clinical and biochemical characteristics of the HFrEF and control groups are presented in Table 1. Twenty-five patients underwent these procedures twice, i.e., at baseline before implantation of CRT device, and after 6 months of the resynchronization therapy. The clinical and biochemical characteristics of this population are presented in Table 2.

The study complies with the 1964 Declaration of Helsinki and its later amendments, and was approved by the institutional medical ethics committee. All the patients were involved in the study after signing an informed written consent for participating in the study, including taking and storage of blood samples.

### 2.2. Echocardiography

Transthoracic echocardiography was based on the two-dimensional measurements and included standard measurements with EF assessment according to the Simpson’s method (biplane method), left ventricular end-diastolic diameter (LVEDD), as well as left ventricular end-diastolic volume (LVEDV) and end-systolic volume (LVESV). Echocardiography was also used to exclude any significant heart abnormalities in the control group. These measurements were performed with the use of an an, with an ultrasound device with the use of a transthoracic probe working with harmonic imaging within the frequency range of 1.6–3.2 MHz (Philips iE33, Bothell, WA, USA).

### 2.3. Cardiopulmonary Exercise Test

The CPET was performed with the use of the Schiller (Baar, Switzerland) CPET equipment using symptom-limited treadmill exercise test with RAMP protocol. The electrocardiogram was continuously monitored for the heart rate, occurrence of ST segment changes and arrhythmias.

### 2.4. Flow Cytometry

Freshly obtained EDTA-anticoagulated whole-blood samples (100 μL) were incubated for 30 min at room temperature with a panel of specific monoclonal antibodies, i.e., anti-CD14 FITC or anti-CD14 PE (clone: MφP9), anti-CD16 PE-Cy5 or anti-CD16 FITC (clone: NKP15), (all from BD PharMingen, Erembodegen, Belgium). Next, red blood cells were lysed for 7 min with FACS Lysing Solution (BD PharMingen, Erembodegen, Belgium), followed by washing with PBS (Corning, NY, USA). Immunostained blood cells were fixed with CytoFix. Flow cytometry analysis was performed with the use of a FACSCalibur flow cytometer (BD Immunocytometry Systems, San Jose, CA, USA) as previously described [14,15]. Appropriate fluorescence minus-one (FMO) controls were applied for setting correct compensations and assuring proper gating. Flow cytometry data were analysed using FlowJo software (TreeStar Inc., Ashland, OR, USA). All samples were blind analysed by two skilled flow cytometry specialists (MT, KG). The applied gating strategy is presented in Figure 1.

Briefly, monocytes were gated based on event morphology (FSC^high^ and SSC^int^) and presence of CD14 (CD14^+^SSC^int^). Next, the Boolean gate was created according to both morphology “AND” CD14^+^ events. Finally, CD14^++^CD16^−^ classical, CD14^++^CD16^+^ intermediate, and CD14^+^CD16^++^ nonclassical monocytes were gated according to the FMO control [16].

Further statistical analysis included absolute values, as well as changes in the 6-month follow-up, which were calculated by subtracting the baseline monocyte frequency from that observed at 6-month follow-up.

### 2.5. Immunoassay

The concentrations of IL-6, sIL-6R, sgp130, sCD163 (R&D Systems, Minneapolis, MN, USA), and soluble tumor necrosis factor-like weak inducer of apoptosis (sTWEAK; eBioscience, Vienna, Austria) were determined using commercially available ELISA kits (R&D Systems, Minneapolis, MN, USA) according to the manufacturer’s instructions. The mean minimum detection limit was 0.039 pg/mL for IL-6 High Sensitivity, 6.5 pg/mL for sIL-6R, 0.08 ng/mL for sgp130, 9.7 pg/mL for sTWEAK, and 0.177 ng/mL for sCD163. ELISA assays recognized both free cytokines and their complexes.

The chemokines’ concentration analysis was conducted in patients treated with CRT with multiplex technique of the commercially available panel of 40 inflammatory chemokines (Human Chemokine Array Q1, RayBiotech Inc., Peachtree Corners, GA, USA), as previously described [17]. This method is based on the specific reaction of antibodies with the chemokine molecules. The visualization of the chemokine–antibody–biotin complex is performed by laser scanner (GenePix 4100A, Axon Instruments Inc., Foster City, CA, USA).

### 2.6. Statistics

Statistical analysis was performed using GraphPad Prism Software (GraphPad Software, San Diego, CA, USA). The distribution of all variables was verified with the Kolmogorov–Smirnov test. Notably, none of the analysed parameters showed normal distribution. Therefore, the Mann–Whitney U test was used to compare the differences between the groups. Additionally, the Wilcoxon test was used to compare the time-course changes in the analysed parameters. The Spearman correlation coefficient with Bonferroni correction was used to assess correlations between the analysed cell subsets, soluble factors, and clinical data. The differences were considered statistically significant at *p* < 0.05. The results are presented as median and interquartile range (IQR).

## 3. Results

First, we analysed the standard biochemical and functional parameters. The control group had better exercise capacity, which was observed in longer 6MWD test, as well as better CPET results, e.g., higher peak oxygen uptake (Table 1). The analysis of biochemical parameters showed higher brain natriuretic peptide (BNP) concentration, while erythrocytes count, haemoglobin concentration and haematocrit were lower in the HFrEF patients (Table 1). Moreover, we found lower iron concentration in the HFrEF patients (Table 1). 

Data are presented as median and interquartile range (IQR) or number and percentage. Statistically significant parameters are marked by italics.

The HFrEF patients were treated with standard pharmacotherapy, including angiotensin-converting enzyme (ACE) inhibitors (80 patients, 94%), mineralocorticoid-receptor antagonists (MRA) (82 patients, 96%), and beta blockers (83 patients, 98%). The clinical and biochemical characteristics of the population treated with CRT is presented in Table 2. The analysis of CRT effectiveness revealed that 25% of patients did not have a reduction in LVESV of at least 15% 6 months after device implantation.

Subsequently, we analysed the frequency of different monocyte subsets in HFrEF patients and controls without HFrEF. We found that the frequency of CD14^+^CD16^++^ nonclassical monocytes was statistically significantly lower in the HFrEF patients compared with the control group (6.98 IQR: 4.95–8.65 vs. 8.37 IQR: 6.47–9.94; *p* = 0.021), while no differences were observed in the frequency of CD14^++^CD16^+^ intermediate (5.2 IQR:4–7.4 vs. 6 IQR: 4.5–7.1; *p =* 0.502) and CD14^++^CD16^∑^ classical (81.4 IQR: 75.6–84.5 vs. 80.6 IQR: 79.3–84.3; *p* = 0.955) subsets.

In addition, a similar frequency of CD14^++^CD16^+^ intermediate monocytes was observed in both the HFrEF without CRT (5.2 IQR: 4–7.4 vs. 6 IQR: 4.5–7.1; *p* = 0.7) and the HFrEF with CRT at baseline vs. the control group (5.1 IQR: 3.7–6.5 vs. 6 IQR: 4.5–7.1; *p* = 0.28) and CD14^++^CD16^∑^ classical monocytes (respectively 81.2 IQR: 75.6–84.5 vs. 80.6 IQR: 79.3–84.3; *p* = 0.68; 81.4 IQR: 78–86.2 vs. 80.6 IQR: 79.3–84.3; *p* = 0.32), while the lower frequency of nonclassical CD14^+^CD16^++^ monocytes was observed in the group of patients with HFrEF without CRT (6.98 IQR: 5.3–8.8 vs. 8.37 IQR: 6.47–9.94; *p* = 0.045) and HFrEF with CRT at baseline compared to the control group (7 IQR: 4.5–8.4, 8.37 IQR: 6.47–9.94; *p* = 0.22) (Figure 2A–C).

Having found a lower frequency of CD14^+^CD16^++^ nonclassical monocytes in the HFrEF patients when compared to the controls without HFrEF at baseline, next we aimed to investigate the effect of CRT on the frequency of analysed monocyte subsets 6 months after CRT device implantation. Interestingly, we observed that applied treatment causes a statistically significant increase in both CD16-expressing monocyte subsets, i.e., CD14^+^CD16^++^ nonclassical (7 IQR: 4.5–8.4 vs. 7.9 IQR: 6.5–9.5; *p* = 0.042) and CD14^++^CD16^+^ intermediate monocytes (5.1 IQR: 3.7–6.5 vs. 6.8 IQR: 5.4–7.4; *p* = 0.017) (Figure 3B,C). Consequently, the frequency of steady-state CD14^++^CD16^∑^ classical monocytes was decreased (81.4 IQR: 78–86.2 vs. 78.2 IQR: 76.1–81.7; *p* = 0.004) (Figure 3A).

The observed shifts in the composition of circulating monocyte subsets brought the frequency of all analysed subsets to the levels found in individuals without HFrEF (Figure 3D,E).

We further set out to analyse whether the observed frequency of different monocyte subsets was associated with the levels of analysed cytokines, cytokine receptors, chemokines and clinical parameters. We found that the baseline frequency of CD14^+^CD16^++^ nonclassical monocytes, CD14^++^CD16^+^ intermediate monocytes, and CD14^++^CD16^−^ classical monocytes correlated with several parameters (Appendix A). An interesting finding was the correlation of CD14^++^CD16^−^ classical monocytes with transferrin saturation (R = −0.343, *p* = 0.002) (Appendix A). Moreover, we found correlations between CRT-induced changes in the frequency of the CD14^+^CD16^++^ nonclassical subset with selected biochemical parameters (Table 3).

## 4. Discussion

Here, we show that the HrREF patients demonstrate lower frequency of CD14^+^CD16^++^ nonclassical monocytes with proinflammatory potential when compared with normal controls without HFrEF. Importantly, we found that CRT in the HFrEF patients causes substantial changes in the composition of circulating monocyte subsets, bringing them to the levels observed in controls without HFrEF. Our observations support the hypothesis that the reversal of cardiac remodelling induced by resynchronization therapy initiates favourable changes in innate immune responses. Moreover, we demonstrated that various chemokine and cytokine levels are associated with monocyte distribution alterations in the HFrEF patients, which are further modified by CRT.

Atherosclerosis is an important diseases contributing to the development of HF, which represents the predominant cause of both acute and chronic cardiovascular diseases, including HFrEF, which belongs to the latter group. Several studies have demonstrated the crucial role of CD16^+^-activated monocytes in atherosclerosis development and progression [6,18]. In addition, the higher frequency of these cells was associated with cardiovascular risk factors and serum concentrations of inflammatory cytokines, including TNF [18]. High numbers or increased activation and migration of CD14^+^CD16^++^ nonclassical monocytes may be associated with unfavourable outcomes, since they were shown to be associated with commonly acknowledged risk factors [19]. Moreover, reduced tissue blood flow, observed in patients with HFrEF, may induce local low-grade inflammation, which in turn may attract circulating monocytes with proinflammatory potential to the local microenvironment. This can explain the lower frequency of CD14^+^CD16^++^ monocytes in the HFrEF patients observed in this study. Moreover, putative increased migration of monocytes with proinflammatory potential may represent a hallmark of HFrEF, regardless of the pathological mechanism of the disease, since no differences in monocyte frequency were observed between the patients qualified and not qualified to receive the CRT treatment. Importantly, however, our data indicate that successful CRT in the HFrEF patients causes the restoration of systemic monocyte composition. This is associated with the improvement in the clinical status, increased tissue blood flow and reduced tissue inflammation [17,20]. More importantly, the increased frequency of CD14^++^CD16^+^ intermediate monocytes after CRT, observed in the present study, may be associated with the induction of natural reparatory mechanisms, since CD14^++^CD16^+^ intermediate monocytes were shown by our and other groups to be involved in the regulation of immune responses, angiogenesis and tissue regeneration [14,15,20].

The pro- or anti-inflammatory properties of monocytes, and consequently the tissue macrophages, are induced on the periphery by growth factors (e.g., G-CSF, GM-CSF), cytokines (e.g., IFN-g, IL-4, IL-10, IL-13, IL-31, TSLP), chemokines (e.g., interferon gamma-induced protein 10-IP-10, CXCL-8, MIP-1a) and small molecules (e.g., lipid mediators, glucocorticoids). Polarization towards proinflammatory cells is induced by proinflammatory mediators, including IFN-γ and IL-31, while anti-inflammatory properties of monocytes and macrophages are stimulated by anti-inflammatory agents, such as IL-10 and glucocorticoids [21]. Activated monocytes directly contribute to this process by secretion of soluble factors. IP-10 is produced by monocytes, among others, and considered a potent chemotactic factor for them. In fact, IP-10 is produced in response to proinflammatory signalling [22]. A recent report by our group members indicated that the HFrEF patients do not present elevated IP-10 serum levels [12]. However, we observed in this study a weak, but statistically significant, positive correlation between the frequency of CD14^+^CD16^++^ nonclassical monocytes and IP-10 serum levels, supporting our findings of proinflammatory monocyte migration to the side of low-grade local inflammation in HFrEF patients. Furthermore, the proinflammatory properties of primed monocytes may also be induced by IL-31 [23]. However, in this study, we found no significant associations with other commonly acknowledged monocyte chemokines in HF, including MCP-1.

The MCP-1-induced motility of normal and leukemic monocytes/macrophages may be controlled by BNP. This mechanism is, however, attenuated in monocytes from HF patients [24]. More importantly, BNP has been shown to inhibit some proinflammatory monocyte/macrophage activities, such as inflammasome activation and associated IL-1β cleavage [25]. This partially explains the positive association of BNP levels and frequency of CD14^++^CD16^+^ monocytes with anti-inflammatory properties observed in this study. Anti-inflammatory properties of macrophages may also be associated with iron metabolism [26].

We found a negative correlation between iron and transferrin saturation and CD14^++^CD16^−^ classical monocytes. Mononuclear phagocytes, including monocytes/macrophages, were shown to play a central role in iron metabolism. Iron uptake by monocytes depends on different processes, i.e., erythrophagocytosis and receptor-dependent processes, such as uptake of haemoglobin and transferrin-bound iron. The correlation observed in the present study suggests the importance of monocytes in iron metabolism. Thus, researchers are searching to understand the regulation of iron transport by cytokines and further by monocytes as a crucial mechanism in the pathogenesis of chronic anaemia [26,27,28].

Previously, researchers have shown that in the course of chronic inflammatory conditions, cytokines induce a diversion of iron traffic, leading to hypoferremia and retention of the metal within the reticuloendothelial system [29]. However, the regulatory pathways underlying these disturbances of iron homeostasis in HF patients are complex and poorly understood. It was shown that proinflammatory, as well as anti-inflammatory molecules, e.g., IL-10, mediate iron uptake into activated monocytes [28]. The intriguing associations between transferrin saturation and analysed monocyte subsets that we observed in this study progress the understanding of the importance of iron supplementation in HFrEF [26,27]. Therefore, the classical monocyte subset may be associated with iron deficiency, which highlights the important association between the inflammatory network and iron homeostasis, which is crucial for understanding the HF pathomechanism.

From the clinical point of view, a better understanding of the implications of inflammatory mediators and immune cells on HF will improve the interpretation of factors contributing to the disease development and progression. One of the best investigated and most important mediators of inflammation is IL-6, which has been found to be elevated in patients with chronic, as well as decompensated HF [17,30]. However, according to the present study and previous reports, the role of IL-6 in the differentiation of monocyte subsets remains unclear [30].

Taken together, we demonstrated that HFrEF is associated with a lower frequency of nonclassical monocytes. However, successful CRT causes beneficial changes in the frequency of classical, intermediate, and nonclassical monocytes, bringing them to the levels observed in controls without HFrEF. Moreover, chemokine and cytokine variations may affect monocyte distribution in the HFrEF patients. The role of different monocyte subsets and the influence of chemokines and cytokines on monocyte phenotype and function in the pathogenesis and progression of chronic heart failure requires further evaluation.

The study was performed on a relatively small HFrEF population (most in NYHA III with severe EF impairment). Thus, the investigation of intragroup correlations was inevitably limited. Moreover, previous studies suggest that the number of circulating monocytes may not always be adequate to their corresponding tissue levels and their actual functional state [31].

## 5. Conclusions

Patients with HFrEF present altered monocyte compositions, associated with decreased frequency of monocytes with high proinflammatory profiles. Successful cardiac resynchronization therapy in patients with HFrEF is associated with changes in the monocyte compartment, making them more similar to the levels observed in controls without HFrEF. In the HFrEF patients, chemokines and cytokines are associated with changes in monocyte distribution, which further translate into patients’ clinical condition. Moreover, classical monocytes may be associated with iron homeostasis.

## Figures and Tables

**Figure 1 cells-10-03482-f001:**
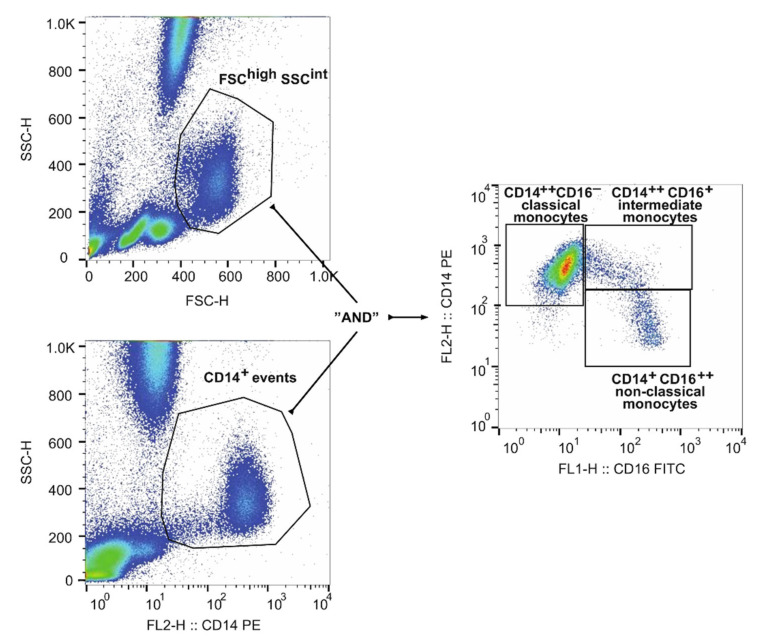
Gating strategy of monocyte analysis with the use of flow cytometry.

**Figure 2 cells-10-03482-f002:**
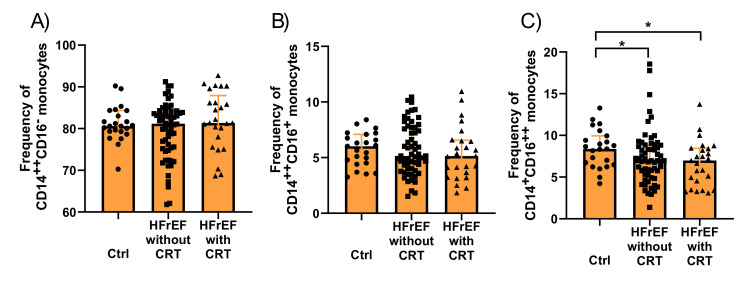
Monocyte subsets in control (Ctrl), patients with heart failure with reduced ejection fraction, who were not qualified for cardiac resynchronization therapy (HFrEF without CRT), and HFrEF patients qualified for cardiac resynchronization therapy (HFrEF with CRT) at baseline. Frequency of (**A**) classical (CD14^++^CD16^−^) (**B**) intermediate (CD14^++^16^+^), and (**C**) nonclassical (CD14^+^16^++^) monocytes. * *p* < 0.05.

**Figure 3 cells-10-03482-f003:**
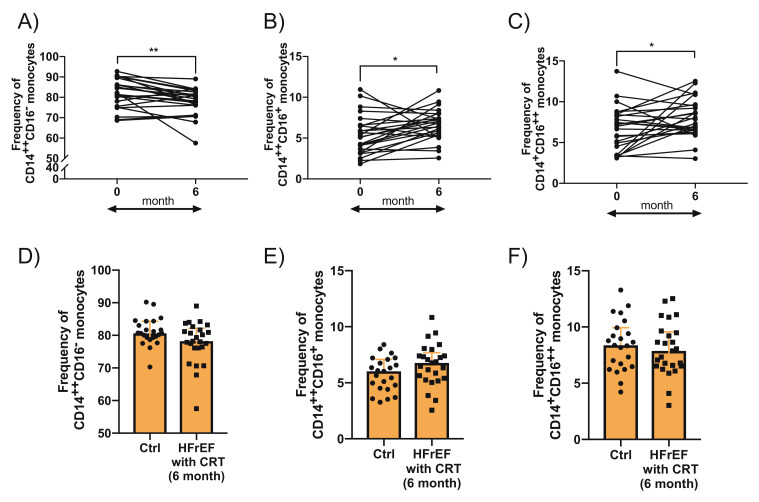
Monocyte subsets in patients with heart failure with reduced ejection fraction (HFrEF) who qualified for cardiac resynchronization therapy (CRT) at baseline (0) and at 6-month follow-up (6). (**A**) classical (CD14^++^CD16^−^) (**B**) intermediate (CD14^++^16^+^), and (**C**) nonclassical (CD14^+^16^++^) monocytes. Monocyte subsets in 6-month follow-up after CRT device implantation compared to controls (**D**) classical (CD14^++^CD16^−^) (**E**) intermediate (CD14^++^16^+^), and (**F**) nonclassical (CD14^+^16^++^) monocytes.* *p* < 0.05; ** *p* < 0.01.

**Table 1 cells-10-03482-t001:** Demographic, functional, echocardiographic, and laboratory characteristics of the heart failure patients with reduced ejection fraction (HFrEF) and control group.

	HFrEF (*n* = 85)	Control Group (*n* = 23)	*p* Value
Age, years	65 (58–72)	62 (56–70)	0.25
Female sex, % (*n*)	12.9 (11)	30.4 (7)	*0.05*
BMI, kg/m^2^	28.2 (26–31.3)	25.3 (23.8–31.2)	0.22
Ischemic heart disease, % (*n*)	63.5 (54)	30.4 (7)	*0.005*
Arterial hypertension, % (*n*)	57.8 (48)	91.3 (21)	*0.003*
Diabetes, % (*n*)	32.5 (27)	30.4 (7)	0.85
**Pharmacotherapy**			
Acetylsalicylic acid, % (*n*)	65.9 (56)	60.9 (14)	0.66
Statin, % (*n*)	78.8 (67)	43.5 (10)	<*0.001*
ACE inhibitor/ARB, % (*n*)	92.9 (79)	82.6 (19)	0.13
**Functional parameters**			
NYHA functional class	3 (2–3)		
6MWD, m	402.5 (305–450)	495 (410–540)	<*0.001*
Peak VO_2_, mL/kg/min	15.3 (12–20.2)	24.1 (19.3–28.8)	<*0.001*
VO_2_ in anaerobic threshold, mL/kg/min,	11.8 (9.9–14)	14.9 (13.3–17.7)	<*0.001*
Peak VCO_2,_ L/min	1.3 (1–1.7)	1.9 (1.7–2.8)	<*0.001*
VE/VCO_2_ slope	31 (27.4–35.3)	27.5 (24.8–30.5)	*0.009*
**Echocardiography**			
EF,%	25 (20–30)	63 (60–69)	<*0.001*
LVEDD, cm	6.6 (5.9–7.2)	4.7 (4.5–5.1)	<*0.001*
LVESV, mL	147 (119–193)	32.5 (28.5–42)	<*0.001*
LVEDV, mL	195 (165–244)	90.5 (77.5–107)	*0.009*
**Laboratory results**			
BNP, pg/mL	179.7 (78.4–426)	25.7 (14.4–41.8)	<*0.001*
CRP, mg/dL	1.8 (1–3.3)	2 (1–2.7)	0.84
Erythrocytes, 10^6^/µL	4.6 (4.3–4.9)	4.9 (4.6–5.2)	*0.004*
Haemoglobin, g/dL	14 (13.3–14.7)	15.1 (14.3–15.6)	<*0.001*
Haematocrit, %	42 (39.2–44)	44.6 (43.1–46.8)	<*0.001*
Leukocytes, 10^3^/µL	6.8 (5.7–7.8)	6.3 (5.7–7.2)	0.25
Platelets, 10^3^/µL	197.5 (167.5–233)	218.5 (198–250)	0.05
Iron, µg/dL	91 (74.6–116)	121 (98–146)	*0.003*
INR	1 (0.95–1.17)	0.93 (0.89–0.96)	*0.007*
Creatinine, mg/dL	1.1 (0.9–1.3)	0.8 (0.76–1)	*0.002*

Abbreviations: 6MWD, 6 min walk distance; ACE inhibitor, angiotensin converting enzyme inhibitor; ARB, angiotensin receptor blocker; BMI, body mass index; BNP, brain natriuretic peptide; CRP, C-reactive protein; EF, ejection fraction; LVEDD, left ventricular end-diastolic diameter; LVEDV, left ventricle end-diastolic volume; LVESV, left ventricle end-systolic volume; VCO_2_, carbon dioxide production; VE/VCO_2_ slope, minute ventilation—carbon dioxide production relationship from the initiation to peak exercise; VO_2_, oxygen uptake; *p*, statistical significance.

**Table 2 cells-10-03482-t002:** Demographic, functional, echocardiographic, and biochemical characteristics of the heart failure patients with implanted CRT at baseline, at 6-month follow-up and control group.

	CRT at Baseline (*n* = 25)	CRT at 6-Month Follow-up (*n* = 25)	*p* Value CRT at Baseline vs. CRT at 6-Month Follow-up	Control Group (*n* = 23)	*p* Value CRT at Baseline vs. Control Group	*p* Value CRT at 6-Month Follow-up vs. Control Group
Age, years	66 (55–73)			62 (56–70)	0.44	0.31
Female sex, % (*n*)	8 (2)	8 (2)		30.4 (7)	*0.05*	*0.05*
**Pharmacotherapy**						
Acetylsalicylic acid, % (*n*)	76 (19)	76 (19)		60.9 (14)	0.26	0.26
Statin, % (*n*)	72 (18)	72 (18)		43.5 (10)	*0.05*	*0.05*
ACE inhibitor/ARB, % (*n*)	88 (22)	88 (22)		82.6 (19)	0.6	0.6
**Functional parameters**						
NYHA functional class	3 (2.5–3)	2 (2–2.5)	<*0.001*			
6MWD, m	390 (320–450)	444 (352.5–480)	*0.031*	495 (410–540)	*0.008*	0.14
Peak VO_2_, mL/kg/min	15.4 (12.7–21.3)	18 (12.2–22.1)	0.309	24.1 (19.3–28.8)	*0.003*	*0.005*
VE/VCO_2_ slope	30.1 (25.6–34.2)	29.1 (25.3–34.1)	0.523	27.5 (24.8–30.5)	0.17	0.27
**Echocardiography**						
EF,%	21 (18–28)	33 (26–39)	<*0.001*	63 (60–69)	<*0.001*	<*0.001*
LVEDD, cm	6.9 (6.3–7.2)	6.3 (5.7–6.8)	*0.002*	4.7 (4.5–5.1)	<*0.001*	<*0.001*
LVESV, mL	176 (138–240)	102.5 (87–162)	*0.001*	32.5 (28.5–42)	<*0.001*	<*0.001*
LVEDV, mL	232 (184–297)	157 (128–253)	*0.002*	90.5 (77.5–107)	<*0.001*	<*0.001*
**Biochemical parameters**						
BNP, pg/mL	168 (88–322.6)	138.9 (32–292.5)	0.19	25.7 (14.4–41.8)	<*0.001*	*0.002*
CRP, mg/dL	1.3 (0.8–2.1)	1.2 (0.8–1.9)	0.831	2 (1–2.7)	0.32	0.1
Erythrocytes, 10^6^/µL	4.5 (4.3–4.8)	4.6 (4.2–4.7)	0.838	4.9 (4.6–5.2)	*0.004*	*0.01*
Haemoglobin, g/dL	14 (13–14.3)	13.9 (13.1–14.2)	1	15.1 (14.3–15.6)	<*0.001*	<*0.001*
Haematocrit, %	41 (39–43.2)	41.9 (39.1–43.4)	0.838	44.6 (43.1–46.8)	*0.001*	*0.004*
Leukocytes, 10^3^/µL	6.9 (5.8–8.2)	6.5 (6–7.8)	0.689	6.3 (5.7–7.2)	0.27	0.29
Platelets, 10^3^/µL	194 (161–237)	184 (151–212)	0.424	218.5 (198–250)	0.09	*0.01*
Iron, µg/dL	92 (79–108)	92 (64–110.5)	0.838	121 (98–146)	*0.01*	*0.004*
INR	1 (0.95–1.1)	1 (0.96–1.1)	0.286	0.93 (0.89–0.96)	*0.02*	*0.003*
Uric acid, mg/dL	7.1 (5.9–7.6)	7 (5.7–8.2)	1	5 (4.7–5.5)	*0.05*	*0.02*
Creatinine, mg/dL	0.9 (0.8–1.3)	1 (0.9–1.3)	0.424	0.8 (0.76–1)	0.08	*0.03*
Urea,mg/dL	47 (38.5–63)	50 (37–57)	0.838	40 (36–41)	*0.03*	0.06
Total cholesterol, mg/dL	164 (147–203)	158 (137.5–170.5)	0.522	201 (179–226)	*0.01*	<*0.001*
HDL cholesterol, mg/dL	43 (36–47)	40 (37–44.5)	0.522	55 (46–60)	<*0.001*	<*0.001*
LDL cholesterol, mg/dL	106 (84–125)	95.5 (75.5–120)	0.838	137 (113–163)	*0.01*	*0.004*

Abbreviations: 6MWD, 6 min walk distance; ACE inhibitor, angiotensin converting enzyme inhibitor; ARB, angiotensin receptor blocker; BMI, body mass index; BNP, brain natriuretic peptide; CRP, C-reactive protein; EF, ejection fraction; LVEDD, left ventricular end-diastolic diameter; LVEDV, left ventricle end-diastolic volume; LVESV, left ventricle end-systolic volume; VE/VCO_2_ slope, minute ventilation carbon dioxide production relationship from the initiation to peak exercise; VO_2_, oxygen uptake; *p*, statistical significance. Data are presented as median and interquartile range (IQR) or number and percentage. Statistically significant parameters are marked by italics.

**Table 3 cells-10-03482-t003:** Correlation of CRT-induced monocyte CD14^+^CD16^++^ nonclassical subset changes with changes in selected biochemical parameters.

	R	*P*
CD14^+^CD16^++^		
sIL-6R	0.497	*0.022*
sgp130	0.434	*0.049*
Ferritin	0.496	*0.019*

Abbreviations: *p*, statistical significance, R, correlation coefficient; sgp130, soluble form of gp130; sIL-6R, soluble IL-6 receptor. Spearman correlation coefficient with Bonferroni correction was used. Statistically significant parameters are marked by italics.

## Data Availability

The data supporting reported results available on request.

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
