# Peer review of "Monocyte Subsets in Patients with Chronic Heart Failure Treated with Cardiac Resynchronization Therapy"

_cells, 2021, doi:10.3390/cells10123482_

Round 1
Reviewer 1 Report
Overall very interesting data but there are a few concerns regarding the manuscript:
Major:
- In order to validate the changes in monocyte subsets in the CRT group, a follow-up of HFrEF patients not receiving a CRT should be performed and changes should be compared to the CRT group
- You are referring in most parts of the conclusion to data which is only available in the supplementary material and has not been mentioned before but is presented as a key result. This should be changed
- I would recommend to avoid causal language as used throughout the conclusion
- Extensive editing of english language is needed as parts of the manuscript can be confusing and difficult to reed
- Correlations for change in LVESV and change in monocyte compartment would be interesting
Minor:
- Figure 2: The marks for level of significance are not where they need to be
- Figure 3: significant difference for D-F should be pointed out as well
Author Response
Major:
- In order to validate the changes in monocyte subsets in the CRT group, a follow-up of HFrEF patients not receiving a CRT should be performed and changes should be compared to the CRT group
Answer: Thank you for making this point. The follow-up of HFrEF patients, who were not qualified for CRT would be of great benefit for this paper. This will need further planned research and will give the better insight into changes of monocyte levels related directly to CRT, not affected by the time of pharmacotherapy. The presented study was designed to assess the changes in monocytes only in patients qualified for CRT as the procedure that changed the treatment conditions of patients with HFrEF, while the remaining patients did not undergo significant changes in HFrEF therapy.
- You are referring in most parts of the conclusion to data which is only available in the supplementary material and has not been mentioned before but is presented as a key result. This should be changed
Answer: Thank you for this comment. We have transferred the data from supplementary materials to manuscript body in the results section as following:
Page 13, line 4:
In addition, a similar frequency of CD14++CD16+ intermediate monocyte was observed in both the HFrEF without CRT (5.2 IQR:4–7.4 vs 6 IQR:4.5–7.1; p = 0.7) and the HFrEF with CRT at baseline vs control group (5.1 IQR: 3.7 – 6.5 vs 6 IQR:4.5–7.1; p = 0.28) and CD14++CD16- classical monocytes (respectively 81.2 IQR:75.6–84.5 vs 80.6 IQR:79.3–84.3; p = 0.68;, 81.4 IQR 78-86.2 vs 80.6 IQR:79.3–84.3; p = 0.32), while the lower frequency of non-classical CD14+CD16++ monocytes was observed in te group of patients with HFrEF without CRT (6.98 IQR 5.3-8.8 vs 8.37 IQR6.47-9.94; p = 0.045) and HFrEF with CRT at baseline compared to control group (7 IQR: 4.5 – 8.4, 8.37 IQR6.47-9.94;, p = 0.22) (Figure 2).
Page 13, line 23: ‘It was found that applied treatment caused a statistically significant increase of both CD16 expressing monocyte subsets, i.e. CD14+CD16++ non-classical (7 IQR: 4.5 – 8.4 vs 7.9 IQR: 6.5 – 9.5; p = 0.042) and CD14++CD16+ intermediate monocytes (5.1 IQR: 3.7 – 6.5 vs 6.8 IQR: 5.4 – 7.4; p = 0.017) (Figure 3 B, C). Consequently, the frequency of steady-state CD14++CD16- classical monocytes was decreased (81.4 IQR 78-86.2 vs 78.2 IQR 76.1-81.7; p = 0.004) (Figure 3 A).
- I would recommend to avoid causal language as used throughout the conclusion
Answer: Thank you for making this point. The Conclusions were changed into: ‘Patients with HFrEF present altered monocyte composition associated with decreased frequency of monocytes with high pro-inflammatory profile. Successful cardiac resynchronization therapy in patients with HFrEF is associated with changes in the monocyte compartment making them similar to the levels observed in controls without HFrEF. In the HFrEF patients, chemokines and cytokines are associated with changes in monocyte distribution, that further translate into patients’ clinical condition. Moreover, classical monocytes may be associated with iron homeostasis.’
’
- Extensive editing of english language is needed as parts of the manuscript can be confusing and difficult to reed
Answer: Thank you for this comment. We have done the English language editing.
- Correlations for change in LVESV and change in monocyte compartment would be interesting.
Answer: Thank you making this point. We have already done this correlations, however it was not significant.
Minor:
- Figure 2: The marks for level of significance are not where they need to be
Answer: Thank you for making this point. We revised the Figure according to your suggestion.
- Figure 3: significant difference for D-F should be pointed out as well
Answer: Thank you for this comment. Let us clarify this point. The levels of significance were not marked because after 6 months of CRT the observed shifts in the composition of circulating monocyte subsets brought frequency of all analyzed subsets to the levels found in normal individuals without HFrEF and were not significant (Figure 3 D-E).

Reviewer 2 Report
The authors examined the changes in monocyte subsets in patients with chronic heart failure treated with cardiac resynchronization therapy. They also compared the monocyte subsets with controls. They found that the rate of non-classical monocytes (activated) was lower in HFrEF patients as compared to control group, while intermediate (activated) and classic (steady state, scavenger) did not differ. After CRT, significant increase of non-classical and intermediate monocytes, while the frequency of steady-state monocytes was decreased. This paper is generally well written, but difficult to read because of the superscript of CD14 or CD16 (++. +, -).
- This is an interesting paper, but control subjects should be carefully selected. In abstract, the authors stated ‘health controls, but control subjects had various underlying diseases. Inappropriate selection of controls might confuse the results. Also, the number of controls is small.
- In Figure 2, why did you present data of patients with heart failure with reduced ejection fraction, who were not qualified for cardiac resynchronization therapy?
Author Response
The authors examined the changes in monocyte subsets in patients with chronic heart failure treated with cardiac resynchronization therapy. They also compared the monocyte subsets with controls. They found that the rate of non-classical monocytes (activated) was lower in HFrEF patients as compared to control group, while intermediate (activated) and classic (steady state, scavenger) did not differ. After CRT, significant increase of non-classical and intermediate monocytes, while the frequency of steady-state monocytes was decreased. This paper is generally well written, but difficult to read because of the superscript of CD14 or CD16 (++. +, -) .
Answer: thank you for this comment. We have removed the superscript of CD14 and CD16 to unify the nomenclature.
- This is an interesting paper, but control subjects should be carefully selected. In abstract, the authors stated ‘health controls, but control subjects had various underlying diseases. Inappropriate selection of controls might confuse the results. Also, the number of controls is small.
Answer: Thank you for this comment. We do agree, that important aspect is to select the control population carefully. The control group consisted of 23 volunteers without the history of HF, with similar median age and body weight, affected with chronic cardiovascular diseases observed in the HFrEF patients. We agree with the Reviewer that the ‘healthy’ controls may be confusing; therefore we have changed it into controls without HFrEF. The limited number of volunteers included into control group results from matching to the group of HFrEF patients treated with CRT (25 patients).
- In Figure 2, why did you present data of patients with heart failure with reduced ejection fraction, who were not qualified for cardiac resynchronization therapy?
Answer: Thank you for this question. We have shown this data for patients with HFrEF who were qualified for CRT and those, who were not, to present that there were no differences in monocyte levels, while both groups present different levels of monocytes when compared to controls.

Round 2
Reviewer 1 Report
Authors changed the manuscript according to recommendations. I advice some further grammar check to optimize the manuscript a little further.
Reviewer 2 Report
The authors have provided acceptable detailed responses to reviewers' critiques and have appropriately revised the manuscript.